# Prognostic factors in severe dengue patients: A multi-center retrospective cohort study

**Cheng-Hsun Yang[1], Ing-Kit Lee[1,2]\*, Yi-Chun Chen[1,2], Wen-Chi Huang[1], Jui-Chi Hsu[1], Chien-Hsiang Tai[1], Chung-Hao Huang[3], Chun-Yu Lin[3], Yen-Hsu Chen[3,4]\***

**1** Division of Infectious Diseases, Department of Internal Medicine, Kaohsiung Chang Gung Memorial Hospital, Kaohsiung, Taiwan (R.O.C.), **2** Chang Gung University College of Medicine, Taoyuan, Taiwan (R. O.C.), **3** Division of Infectious Diseases, Department of Internal Medicine, Kaohsiung Medical University Hospital, School of Medicine, College of Medicine, Center for Tropical Medicine and Infectious Disease Research, Kaohsiung Medical University, Kaohsiung, Taiwan (R.O.C.), **4** School of Medicine, College of Medicine, National Sun Yat-Sen University, Kaohsiung, Taiwan (R.O.C.)

\* leee@cgmh.org.tw (IKL); infchen@gmail.com (YHC)

## Abstract

### Background/purpose

Early detection of severe dengue (SD) and appropriate management are crucial in reducing the case fatality rate. The objective of this study was to investigate the clinical characteristics of SD and identify independent risk factors associated with mortality among SD patients.

### Methods

A retrospective study was conducted at two medical center hospitals between 2002 and 2019, involving patients aged ≧18 years with laboratory-confirmed SD.

### Results

This study included 294 patients with SD, of whom 203 (69%) survived and 91 (31%) died. Among the 294 SD patients, 103 (35%) experienced acute kidney injury, 54 (18.4%) had pneumonia, and 19 (6.5%) had bacteremia. Among the 286 patients with available alanine aminotransferase (ALT) data, 41 (14.3%) experienced severe hepatitis (ALT>1000U/L). The median time from illness onset to death among the 91 SD patients who died was 5 days. Multivariable regression analysis revealed increasing odds of death associated with older age (odds ratio [OR], 1.037; 95% confidence interval [CI], 1.009–1.066), altered consciousness (OR, 8.591; 95% CI, 2.914–25.330), gastrointestinal bleeding (OR, 1.939; 95% CI, 1.037–3.626), and leukocytosis (OR, 2.504; 95% CI, 1.124–5.578) upon arrival, as well as organ impairment during hospitalization, including acute kidney injury (OR, 2.627; 95% CI, 1.373–5.028), severe hepatitis (OR, 5.324; 95% CI, 2.199–12.889), and pneumonia (OR, 2.250; 95% CI, 1.054–4.802).

**Data Availability Statement:** All relevant data are within the manuscript and its Supporting Information files.

**Funding:** This work was supported by grants from Kaohsiung Gang Gung Memorial Hospital, Kaohsiung, Taiwan (R.O.C.) (no. CMRPG8M1532 awarded to IKL) and National Science and Technology Council, Taiwan (no.113-2314-B-110-003 awarded to YHC). The funders had no role in study design, data collection and analysis, decision to publish, or preparation of the manuscript.

**Competing interests:** The authors have declared that no competing interests exist.

## Conclusions

Our findings underscore the importance of early recognition and intervention by frontline physicians in identifying SD patients at high risk of mortality. This information can significantly contribute to reducing fatalities and improving the overall management of SD cases.

## Author summary

The early identification of severe dengue (SD) patients who are at high risk of mortality is crucial for frontline physicians to deliver timely and intensive clinical interventions. This study is particularly significant as it focuses on SD patients and explores the factors associated with mortality, aiming to reduce preventable deaths within this specific patient group. We have pinpointed several independent risk factors that are predictive of mortality outcomes, including advanced age, altered consciousness, elevated leukocyte counts, and gastrointestinal bleeding upon admission. Addressing these risk factors at the initial presentation is critical for mitigating the severity of SD and decreasing mortality rates. Additionally, the development of acute kidney injury, severe hepatitis, and pneumonia during hospitalization has been identified as significant prognostic indicators. Prompt recognition of signs of organ hypoperfusion and the implementation of intensive supportive management to prevent further deterioration are essential strategies to reduce mortality among SD patients. These insights are particularly vital for regions with limited medical resources, enabling healthcare providers to efficiently identify and prioritize SD patients at increased risk of mortality.

## Introduction

Dengue, the most prominent mosquito-borne arboviral disease, has witnessed an increase in its incidence over the past fifty years [1]. Globally, an estimated 3.9 billion people from over 128 countries are at risk of dengue virus (DENV) infection, with 284–528 million cases occurring annually, out of which 96 million manifest clinically as severe form of dengue [2,3]. Dengue is caused by four genetically related but antigenically distinct serotypes of the DENV (DENV-1, DENV-2, DENV-3, and DENV-4) [4]. Infection provides life-long immunity to the homologous serotype but only transient protection against the other three serotypes [5]. Secondary dengue infection significantly increases the risk of severe clinical outcomes such as dengue hemorrhagic fever and dengue shock syndrome, primarily due to antibody-dependent enhancement. [6]. In Taiwan, all four serotypes have been detected, with DENV-1 and DENV-2 being predominant, especially during the 2014–2015 epidemic [7].

DENV infection can result in a spectrum of clinical manifestations, ranging from asymptomatic or mild febrile illness to severe life-threatening conditions such as dengue hemorrhagic fever and dengue shock syndrome that can be fatal [8]. In 2009, the World Health Organization (WHO) guidelines classified dengue into dengue with/without warning signs and severe dengue (SD) [8]. SD encompasses cases that include any of the following: (1) plasma leakage leading to shock or respiratory distress, (2) severe bleeding, or (3) organ failure (e.g., elevated liver enzyme levels, impaired consciousness, or heart failure). Factors such as secondary dengue infections, age, viral load, and the infecting serotype and genotype have been implicated in the progression of SD [9–11]. Notably, the outbreak of DENV-2 in Brazil was associated with heightened disease severity and a significantly elevated mortality rate [12].

Despite several proposed risk factors for the progression of SD [9–12], our understanding of the pathogenesis of SD remains limited. It is crucial for frontline physicians to overcome the significant challenge posed by dengue, which involves early recognition and timely management of SD patients, particularly during outbreaks. This challenge arises due to delays and difficulties in differentiating SD individuals who are at a higher risk of fatal outcomes. Therefore, the objective of this study was to identify an independent risk factor for fatality among SD patients, which could aid frontline physicians in early identification of those at high risk and enable timely management to reduce the mortality rate.

## Methods

### Ethics statement

This study was approved by the Institutional Review Board of Kaohsiung Chang Gung Memorial Hospital (Document no. 202300967B0) and Kaohsiung Medical University Memorial Hospital (Document no. KMUHIRB-G(I)-20190048). All research was conducted in accordance with both the Declarations of Helsinki and Istanbul. Informed consent was not required in view of the retrospective study design, and all data were de-identified prior to analysis.

### Patients and setting

We conducted a retrospective study that included eligible patients aged 18 years and older with laboratory-confirmed dengue between 2002 and 2019. These patients received treatment at Kaohsiung Chang Gung Memorial Hospital (2,700 beds) and Kaohsiung Medical University Memorial Hospital (1,700 beds), both of which serve as primary care and tertiary referral centers in Taiwan. These hospitals provide comprehensive services, including outpatient and inpatient care, as well as a 24-hour emergency department for patients requiring immediate medical attention. The inclusion criteria for this study were dengue patients who exhibited SD, as defined by the 2009 WHO criteria [8]. Patients with non-severe dengue, both with and without warning signs, were excluded from the study. The data used in this study were primarily sourced from electronic medical records of the hospitals and supplemented by a secondary manual search. The recorded data included demographic characteristics, interval between illness onset and presentation, mortality information, DENV serotype, symptoms/signs, laboratory features at hospital presentation, laboratory test results, complications throughout the clinical course, and in-hospital mortality. Additionally, the warning signs outlined by the WHO in 2009 that manifested in SD patients were recorded for analysis.

All dengue cases included in this study were confirmed using at least one of the following criteria [13]: (i) positive DENV-specific real-time reverse transcription polymerase chain reaction (RT-PCR) using the QuantiTect SYBR Green RT-PCR kit (Qiagen, Hilden, Germany), (ii) positive DENV-specific immunoglobulin (Ig) M antibody (SD DENGUE IgM CAPTURE enzyme-linked immunosorbent assay) in acute-phase serum, (iii) a fourfold increase in DENV-specific IgG antibody (SD DENGUE IgG CAPTURE enzyme-linked immunosorbent assay) in convalescent serum compared to the acute phase, or (iv) positive DENV-specific non-structural glycoprotein-1 antigen in acute-phase serum using the kit from Bio-Rad Laboratories (Marnes-la-Coquette, France), AsiaGen Corp (Taiwan), or SD BIOLINE (Korea).

The DENV serotype was determined using serotype-specific RT-PCR, with the primer sequences for detecting and serotyping the DENV described elsewhere [14,15]. For example, the primer sequences for detecting DENV-2 were as follows: forward primer 5′–GGCTTA GCGCTCACATCCA–3′, reverse primer 5′–GCTGGCCACCCTCTCTTCTT–3′, and nested fluorescent probe sequence FAM-5′–CGCCCACCACTATAGCTGCCGGA–3′-TAMRA [15]. The detection limits for DENV serotypes are as follows: DENV-1 primers can detect as low as

10 PFU/mL of DENV-1; DENV-2 primers can detect as low as 4.6 PFU/mL of DENV-2; DENV-3 primers can detect as low as 4.1 PFU/mL of DENV-3; and DENV-4 primers can detect as low as 5 PFU/mL of DENV-4. The dengue diagnostic tests were conducted by the Centers for Disease Control and Prevention in Taiwan.

### Definitions

The warning signs, as defined by the 2009 WHO, include abdominal pain or tenderness, persistent vomiting, clinical fluid accumulation, mucosal bleeding, and lethargy/restlessness [8]. However, we were unable to include hepatomegaly and an increase in hematocrit concurrent with a rapid decrease in platelet count as warning signs in our analysis due to the lack of information. SD was defined based on the 2009 WHO criteria, which categorizes it as dengue with any of the following symptoms/signs found in group C: severe plasma leakage leading to shock or fluid accumulation with respiratory distress, severe bleeding, or severe organ impairment such as elevated transaminases ≥1,000 IU/L or impaired consciousness [8]. In our analysis, severe hepatitis was defined as serum alanine aminotransferase (ALT) levels exceeding 1,000 U/L (reference value, <40 U/L), consistent with the definition of SD according to the 2009 WHO dengue criteria [8,16]. Gastrointestinal bleeding was identified by the presence of hematemesis and/or the passage of tarry or bloody stool. Acute kidney injury was defined as a rapid increase in the serum creatinine level of >0.5 mg/dL compared to the level at hospital presentation.

### Statistical analysis

To analyze the predictors of mortality among SD patients, we compared survivors and non-survivors based on demographic characteristics, clinical features, laboratory findings, and complications. The normality of data variables was assessed using the Kolmogorov–Smirnov test, which identified hematocrit as the only variable following a normal distribution. Continuous variables were analyzed using the Student's t-test or the Mann–Whitney U test, with results presented as median and range. Categorical variables were compared using the $\chi^2$ test or Fisher's exact test and are reported as frequency and percentage. A two-tailed P-value of < 0.05 was considered statistically significant. The variables that showed significance in the univariate analyses were included in a multivariate logistic regression model to determine independent predictor(s) of mortality among SD patients. All statistical analyses were performed using SPSS version 25 software (Chicago, Armonk, NY, USA).

## Results

### Patient characteristics

During the study period, a total of 294 SD patients were included, comprising 168 men and 126 women, with a median age of 67 years. Among them, 203 (69%) survived while 91 (31%) died. The median time from illness onset to hospital presentation was 2 days (range, 1–22 days) for the 294 SD patients. DENV-2 was identified in 93% of the patients included in the study. The most common comorbidities among the 294 SD patients were hypertension (63%), diabetes mellitus (39%), and chronic kidney disease (20%). Fever was the most frequently observed symptom, reported by 77% of the patients, followed by myalgia (21%) and bone pain (19%). Regarding the warning signs, gastrointestinal bleeding was present in 129 patients (43%), abdominal pain in 45 patients (15.3%), vomiting in 43 patients (14.6%), and altered consciousness in 36 patients (12.2%). Pleural effusion was observed in 51 out of 276 patients with available data, accounting for 18.5% of the cases. Among the 294 SD patients, 103 patients

(35%) experienced acute kidney injury, 54 patients (18.4%) had pneumonia, and 19 patients (6.5%) had bacteremia. Severe hepatitis, as indicated by elevated ALT levels, was reported in 41 out of 286 patients with available data, accounting for 14.3% of the cases. Table 1 provides a detailed overview of the characteristics of the patients included in the study.

## Characteristics of the 91 fatal patients

Among the 91 fatal cases, the median age was 70 years (range: 33–91), with 47 (51.6%) being males. Among these cases, 59 (64.8%) had hypertension, 43 (47%) had diabetes mellitus, and 23 (25.2%) had chronic kidney disease. The median time from the onset of illness to hospital presentation was 2 days (range: 1–10 days), while the median time from onset to fatality was 5 days (range: 1–47 days). The three most common warning signs among the deceased cases were gastrointestinal bleeding (54.9%), altered consciousness (26.4%), and pleural effusion (23%). Among the 91 deceased patients, acute kidney injury was found in 54 (59.3%) patients, severe hepatitis in 30 (33%), and pneumonia in 22 (24.4%).

## Comparison between survivors and non-survivors (Tables 1 and 2)

Non-survivors presented to the hospital significantly earlier after illness onset compared to survivors. Cough, diarrhea, drowsiness, and gastrointestinal bleeding were reported significantly more frequently in non-survivors (28.6%, 25.3%, 26.4%, and 54.9%, respectively) than in survivors (11.8%, 13.8%, 5.9%, and 38.9%, respectively) (Fig 1). On the other hand, skin rash was significantly less reported in non-survivors (5.5%) compared to survivors (14.8%) (Fig 1). Non-survivors also had a significantly higher incidence of leukocytosis at presentation (23% vs. 11.2%) (Fig 2A). Furthermore, significantly higher levels of aspartate transaminase, ALT, and lactate concentrations, as well as lower platelet count throughout the hospitalization, were observed in non-survivors (2538 U/L, 1014 U/L, 29.8 mg/dL, and 13 x $10^9$ cells/L, respectively) compared to survivors (129 U/L, 85 U/L, 13.8 mg/dL, and 16.6 x $10^9$ cells/L, respectively) (Fig 2B–2E). Non-survivors also had a significantly higher incidence of acute kidney injury and severe hepatitis (59.3% and 33%, respectively) compared to survivors (24.1% and 5.6%, respectively).

The results of the multivariate analysis revealed several independent risk factors for mortality in SD patients (Table 2). These included older age (odds ratio [OR], 1.037; 95% confidence interval [CI], 1.009–1.066; P = 0.009), presence of altered consciousness (OR, 8.591; 95% CI, 2.914–25.330; P = 0.000), and gastrointestinal bleeding (OR, 1.939; 95% CI, 1.037–3.626; P = 0.038) upon arrival. Furthermore, leukocytosis at initial presentation (OR, 2.504; 95% CI, 1.124–5.578; P = 0.025), as well as organ involvement during the hospital stay, including the development of acute kidney injury (OR, 2.627; 95% CI, 1.373–5.028; P = 0.004), severe hepatitis (OR, 5.324; 95% CI, 2.199–12.889; P = 0.000), and pneumonia (OR, 2.250; 95% CI, 1.054–4.802; P = 0.036), were also identified as significant risk factors.

## Discussion

Early identification of SD patients who are at high risk of mortality is crucial for frontline physicians to provide timely and aggressive clinical management. This study is highly significant as it focuses on patients with SD and investigates the risk factors associated with mortality, with the aim of reducing preventable deaths in this specific patient group. In this multi-center cohort study, we made significant findings regarding prognostic factors among patients with SD. Specifically, we identified several independent risk factors that can aid in predicting mortality outcomes. These include elderly age, altered consciousness, leukocytosis, and gastrointestinal bleeding upon arrival. Additionally, the development of acute kidney injury, severe

**Table 1. Characteristics of patients with severe dengue.**

| Variable | Overall (N = 294) | Non-survivor (n = 91) | Survivor (n = 203) | P |
|---|---|---|---|---|
| Median age (range), years | 67 (2–91) | 70 (33–91) | 66 (2–88) | 0.092 |
| Age > 65 years | 201 (68) | 65 (71) | 136 (67) | 0.450 |
| Male | 168 (57) | 47 (51.6) | 121 (59.6) | 0.202 |
| Comorbidity | | | | |
| Diabetes mellitus | 117 (39) | 43 (47) | 74 (36.4) | 0.080 |
| Hypertension | 188 (63) | 59 (64.8) | 129 (63.5) | 0.832 |
| Chronic kidney disease | 60 (20.4) | 23 (25.2) | 37 (18.2) | 0.166 |
| End stage renal disease | 21 (7) | 8 (8.7) | 13 (6.4) | 0.462 |
| Ischemic heart disease | 41 (14) | 15 (16.4) | 26 (12.8) | 0.400 |
| Median time from illness onset to hospital presentation (range), day | 2 (1–22) | 2 (1–10) | 3 (1–22) | **0.001** |
| Median time from illness onset to fatality (range), day | - | 5 (1–47) | - | - |
| Dengue virus serotype | | | | 0.072 |
| Serotype 1 | 6 (2) | 3 (3.3) | 3 (1.5) | |
| Serotype 2 | 273 (92.8) | 87 (95.6) | 186 (91.6) | |
| Serotype 3 | 15 (5.1) | 1 (1.1) | 14 (6.9) | |
| **Symptoms and signs upon arrival** | | | | |
| Fever | 227 (77.2) | 72 (79.1) | 155 (76.4) | 0.601 |
| Myalgia | 63 (21.4) | 21 (23) | 42 (20.7) | 0.645 |
| Bone pain | 56 (19) | 18 (19.8) | 38 (18.7) | 0.830 |
| Rash | 35 (11.9) | 5 (5.5) | 30 (14.8) | **0.023** |
| Headache | 52 (17.6) | 18 (19.8) | 34 (16.7) | 0.529 |
| Cough | 50 (17) | 26 (28.6) | 24 (11.8) | **<0.001** |
| Retro-orbital pain | 7 (2.4) | 2 (2.2) | 5 (2.5) | 0.890 |
| Diarrhea | 51 (17.3) | 23 (25.3) | 28 (13.8) | **0.016** |
| Petechiae | 33 (11.2) | 12 (13.2) | 21 (10.3) | 0.475 |
| **Warning signs upon arrival** | | | | |
| Abdomen pain | 45 (15.3) | 16 (17.6) | 29 (14.3) | 0.468 |
| Vomiting | 43 (14.6) | 14 (15.4) | 29 (14.3) | 0.844 |
| Altered consciousness | 36 (12.2) | 24 (26.4) | 12 (5.9) | **<0.001** |
| Mucosal bleed | | | | |
| Gastrointestinal bleed | 129 (43.9) | 50 (54.9) | 79 (38.9) | **0.010** |
| Hemoptysis | 9 (3.1) | 2 (2.2) | 7 (3.5) | 0.472 |
| Gum bleed | 14 (4.76) | 5 (5.5) | 9 (4.4) | 0.693 |
| Clinical fluid accumulation, no./total no. (%) | | | | |
| Pleural effusion | 51/276 (18.5) | 21/91 (23) | 30/185 (16.2) | 0.167 |
| Ascites | 20/256 (7.8) | 7/85 (8.2) | 13/171 (7.6) | 0.859 |
| **Laboratory results upon arrival** | | | | |
| Median WBC (range) ($\times10^9$ cells/L) | 5.9 (0–45.9) (n = 279) | 7.0 (0.5–45.9) (n = 85) | 5.5 (0–24.9) (n = 174) | 0.073 |
| Leukocytosis (WBC >$10\times10^9$ cells/L), no./total no. (%) | 42/279 (15) | 24/91 (23) | 21/188 (11.2) | **0.001** |
| Median hemoglobin (range) (g/dL) | 12.8 (4.2–19.9) (n = 277) | 12.8 (6.5–18.1) (n = 89) | 12.8 (4.26–19.9) (n = 186) | 0.534 |
| Median hematocrit (range) (%) | 38 (12.6–64.5) (n = 275) | 38 (13–54.9) (n = 89) | 37 (12.6–64.5) (n = 186) | 0.720 |
| Median platelet count (range) ($\times10^9$ cells/L) | 86.8 (0.3–501) (n = 280) | 69 (0.3–378) (n = 91) | 71 (0.7–501) (n = 189) | 0.338 |
| Platelet count < 100 $\times10^9$ cells/L, no./total no. (%) | 174/280 (62.1) | 57/91 (62.6) | 117/189 (62) | 0.732 |
| Platelet count < 50 $\times10^9$ cells/L, no./total no. (%) | 119/280 (42.5) | 40/91 (44) | 79/189 (41.8) | 0.833 |
| Median AST (range) (IU/L) | 141.5 (14–16001) (n = 244) | 143 (16–16001) (n = 77) | 141 (14–16001) (n = 167) | 0.271 |
| Median ALT (range) (IU/L) | 418.5 (10–6366) (n = 251) | 85 (10–6366) (n = 78) | 77 (10–3348) (n = 173) | 0.150 |
| Median CRP (range) (mg/L) | 51.4 (0.7–380.8) (n = 120) | 35.4 (3.6–380.8) (n = 44) | 28.7 (0.7–265.2) (n = 76) | 0.230 |

*(Continued)*

**Table 1.** (Continued)

| Variable | Overall (N = 294) | Non-survivor (n = 91) | Survivor (n = 203) | P |
|---|---|---|---|---|
| **Laboratory results during hospitalization** | | | | |
| Median WBC (range) (×10⁹ cells/L) | 13.5 (3.1–72.7) (n = 259) | 14.8 (4.1–34.2) (n = 85) | 13.4 (3.1–72.7) (n = 174) | 0.318 |
| Median hematocrit (range) (%) | 40.1 (23–64.5) (n = 275) | 40.8 (23–59) (n = 89) | 39.9 (25.2–64.5) (n = 186) | 0.652 |
| Median platelet count (range) (×10⁹ cells/L) | 39.6 (3–376) (n = 278) | 13 (3–303) (n = 90) | 16.6 (3–376) (n = 188) | **0.027** |
| Median AST (range) (U/L) | 1933.3 (10–36981) (n = 243) | 2538 (14–36981) (n = 78) | 129 (10–16001) (n = 165) | **<0.001** |
| Median ALT (range) (U/L) | 722.5 (10–19347) (n = 240) | 1014 (10–19347) (n = 75) | 85 (10–3348) (n = 165) | **<0.001** |
| Median troponin-I (range) (ng/mL) | 0.3 (0.01–80) (n = 88) | 0.3 (0.01–80) (n = 50) | 0.08 (0.01–464) (n = 38) | 0.303 |
| Median CPK (range) (U/L) | 473 (19–31737) (n = 57) | 627.5 (49–31737) (n = 30) | 364 (19–103420) (n = 27) | 0.677 |
| Median lactate (range) (mg/dL) | 25.8 (4.2–243) (n = 44) | 29.8 (4.2–243) (n = 35) | 13.8 (8.1–35.2) (n = 9) | **0.012** |
| **Complications during hospitalization** | | | | |
| Acute kidney injury | 103 (35) | 54 (59.3) | 49 (24.1) | **<0.001** |
| Severe hepatitis (ALT >1000 IUL), no./total no. (%) | 41/286 (14.3) | 30/91 (33) | 11/195 (5.6) | **<0.001** |
| Bacteremia | 19 (6.5) | 10 (11) | 9 (4.4) | 0.062 |
| Pneumonia | 54 (18.4) | 22 (24.4) | 32 (15.7) | 0.077 |

Data are presented as number (%) unless otherwise indicated.

The statistically significant variables are highlighted in bold.

ALT = alanine aminotransferase; AST = aspartate aminotransferase; CPK = creatine phosphokinase; CRP = C reactive protein; no./total no. = number of cases/number of overall cases with data available; WBC = white blood cell count.

hepatitis, and pneumonia during hospitalization were also identified as important prognostic indicators. These findings hold particular importance for regions with limited medical resources. By recognizing these risk factors early on, healthcare professionals can proactively identify SD patients who are at a high risk of mortality. This information allows for timely interventions and appropriate allocation of limited resources, potentially preventing the worsening of their condition and improving patient outcomes.

The mortality rates for dengue patients have been investigated in various studies. A study conducted in New Delhi reported that 12 out of 198 patients admitted to the intensive care unit with dengue died, resulting in a mortality rate of 6.1% [17]. Similarly, a study conducted in Brazil involving 97 adult patients with dengue admitted to the intensive care unit reported an in-hospital mortality rate of 19.6% [18]. In Taiwan, a study involving 143 dengue patients admitted to the intensive care unit reported that 33 of these patients died, resulting in a mortality rate of 23.1% [19]. In present study, we found that the fatality rate among SD patients is as high as 31%, which is higher than the rates reported in other studies. Our study was a

**Table 2. Multivariate analysis of independent risk factors associated with fatality in severe dengue patients.**

| Variable | Odds ratio | 95% Confidence Interval | P |
|---|---|---|---|
| Old age | 1.037 | 1.009–1.066 | 0.009 |
| Symptom/sign and laboratory data, assessed upon arrival | | | |
| Altered consciousness | 8.591 | 2.917–25.330 | <0.001 |
| Gastrointestinal bleeding | 1.939 | 1.037–3.626 | 0.038 |
| Leukocytosis | 2.504 | 1.124–5.578 | 0.025 |
| Occurrence of organ impairment during hospitalization | | | |
| Acute kidney injury | 2.627 | 1.373–5.028 | 0.004 |
| Severe hepatitis | 5.324 | 2.199–12.889 | <0.001 |
| Pneumonia | 2.250 | 1.054–4.802 | 0.036 |

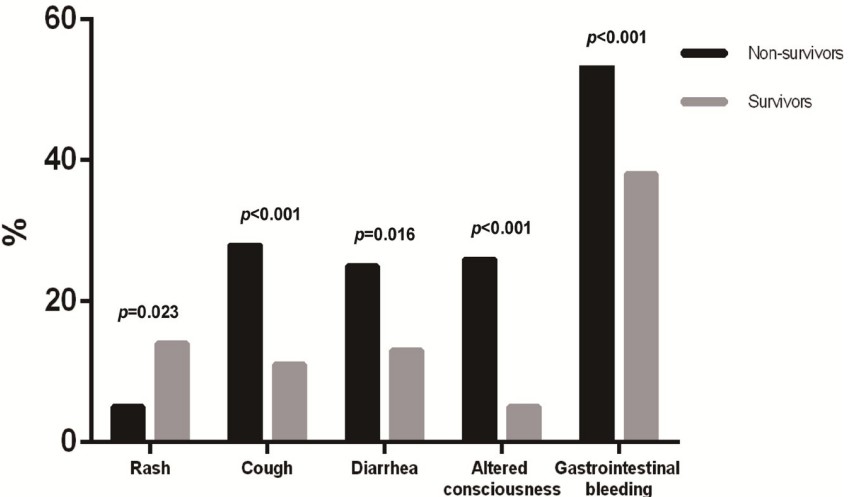

**Fig 1. Significant symptoms and signs at hospital presentation in survivors versus non-survivors.**

multicenter study that enrolled a larger number of cases with SD than previous studies. Furthermore, our study focused specifically on patients classified as group C according to the 2009 WHO definition [8], which accurately captures the true severity of the clinical scenario. This deliberate inclusion criterion may explain why our study outcomes demonstrated worse outcomes compared to previous studies. Of particular note is the finding that the median time from the onset of illness to fatality was 5 days. This underscores the urgency and time-sensitive nature of SD management. Our findings highlight the crucial role of frontline physicians in promptly detecting patients with SD who are at risk of death and providing them with appropriate management to prevent otherwise avoidable mortality and improve patient outcomes.

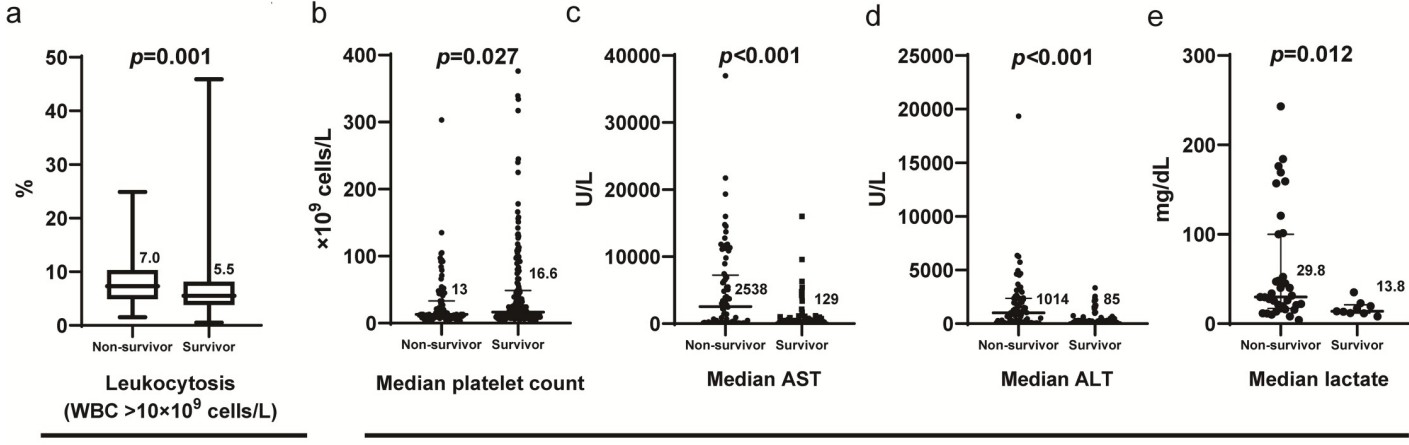

**Fig 2.** Significant laboratory findings at hospital presentation and throughout hospitalization in survivors versus non-survivors: (a) Presence of leukocytosis at hospital presentation, (b) median platelet count during hospitalization, (c) median AST levels during hospitalization, (d) median ALT levels during hospitalization, and (e) median lactate levels during hospitalization. ALT = alanine aminotransferase; AST = aspartate aminotransferase.

DENV infection is increasingly affecting elderly patients worldwide, and our study confirmed that this population experiences worse clinical outcomes, as previously reported [11,20,21]. In regions where dengue is endemic, a noteworthy proportion of patients acquired past dengue infections as they advanced in age. This historical exposure to the DENV further amplified their susceptibility to developing SD [22]. Furthermore, the higher rates of SD observed in elderly patients may be attributed to their co-existing medical conditions. Hypertension and diabetes, both recognized as risk factors for SD, were more prevalent among the elderly [23]. Additionally, aging itself results in immune dysregulation, characterized by defects in T and B cell function, as well as impaired cytokine response [24]. These immunological changes further contribute to the risk of developing SD and mortality in this aging population [25]. Our findings emphasize the significance of clinicians maintaining vigilance when treating elderly patients with dengue, considering the higher frequency of these cases and the elevated risk of mortality. It is crucial to acknowledge the distinct challenges and specific risk factors linked to dengue in the elderly in order to deliver timely and appropriate management, thus reducing adverse outcomes in this vulnerable population.

The WHO has documented that mucosal bleeding and the presence of lethargy or restlessness are pivotal warning signs for the development of SD [8]. Our study illustrates that altered consciousness and gastrointestinal bleeding observed upon arrival not only serve as crucial warning signs for SD but also emerge as early predictors of an unfavorable outcome in dengue patients affected by this condition. Previous studies have reported an incidence of neurological manifestations in dengue patients ranging from 1% to 5%, with a mortality rate of 5% to 30% in patients who develop neurological complications [26,27]. The cause of altered consciousness in dengue patients could be multifactorial, attributed to the direct invasion of the virus or indirect effects resulting from electrolyte imbalance, organ failure, and hemorrhage [26–28]. Moreover, severe plasma leakage in SD may lead to reduced cerebral perfusion, compromising the proper functioning of the brain, especially in susceptible populations such as elderly patients with multiple comorbidities [29]. It is worth noting that aging was also documented as a predictor for a poor outcome in this study, further emphasizing the significance of the impaired state of consciousness in dengue patients.

The presence of gastrointestinal bleeding upon arrival in dengue is an important early prognostic factor for patients with SD. Assir et al. reported that more than 50% of deceased dengue patients experienced some form of bleeding manifestation [30]. Another study series, which included 30 fatal dengue cases, revealed that 80% of the fatal patients experienced gastrointestinal bleeding, with severe hemorrhage accompanied by shock accounting for 30% of fatalities [31]. Gastrointestinal bleeding can lead to hypovolemia, circulatory collapse, and shock. Delayed and inadequate fluid replacement, as well as insufficient transfusion of blood products in cases of severe bleeding, contribute to the development of circulatory failure and increased mortality [32]. Our findings emphasize the critical importance of early identification of gastrointestinal bleeding in SD patients, as it can help timely identify those at higher risk of mortality and enable prompt management accordingly.

In our study, we found a significant association between leukocytosis at the time of presentation and mortality in patients with SD. SD patients with leukocytosis had a more than twofold higher risk of death. While leukopenia is commonly observed in dengue, the presence of leukocytosis in SD patients indicates an unfavorable outcome. This finding is consistent with a study by Nelson et al., where they observed leukocytosis in 67% of shock cases and 66% of mortality cases among patients with dengue hemorrhagic fever [33]. Another study involving 143 critically ill dengue patients revealed that 80 (55.9%) of them developed bacterial infections, with 13 (39.4%) experiencing bacteremia [19]. The presence of leukocytosis upon hospital presentation in SD patients can be attributed to significant plasma leakage, with or without

early organ hypoperfusion and/or early bacterial co-infection. In our series, although not statistically significant, we observed higher hematocrit and C-reactive protein levels at hospital presentation in non-surviving SD patients compared to survivors. Furthermore, there was a significant high incidence of gastrointestinal bleeding at presentation in non-surviving SD patients, which may exacerbate existing plasma leakage and contribute to the development of leukocytosis. This finding carries particular importance in remote areas where laboratory facilities are limited. Identifying high-risk SD patients and referring them promptly to higher-level medical centers before the onset of unfavorable complications can play a crucial role in mitigating the impact of the disease.

The presence of organ impairment during hospitalization is another crucial factor contributing to mortality in patients with SD [13,34]. In our study, we found that the development of severe hepatitis in patients with SD was associated with a fivefold increase in the risk of death. Furthermore, acute kidney injury and pneumonia were identified as factors contributing to a twofold higher risk of mortality compared to those without these complications. The DENV is hepatotropic and known to cause hepatitis of varying severity, occasionally leading to massive hepatic necrosis [35]. Woon et al. demonstrated severe liver involvement in over 60% of fatal cases [36]. Furthermore, Khalil et al. identified acute kidney injury in 13.3% of a series of 532 patients with DENV infection [37]. The development of acute kidney injury was associated with longer hospital stays and higher mortality rates [38–40]. In our series, a higher incidence of bacteremia was observed in deceased SD patients compared to survivors (11% versus 4.4%), although this difference was not statistically significant. Prolonged hypotension and shock resulting from severe plasma leakage and/or bacterial co-infection in cases of SD could lead to organ impairment such as liver and kidney damage. Moreover, the development of pneumonia indicates nosocomial infection, which can exacerbate the dengue disease and contribute to fatal outcomes [41,42]. The importance of our study findings is to underscore the early detection of organ impairment in SD patients and the timely administration of adequate fluid replacement as vital management strategies. Additionally, empirical antibiotic treatment is necessary for elderly SD patients with altered consciousness and leukocytosis at hospital presentation when bacterial co-infection cannot be excluded.

There are several limitations of this study that need to be addressed. Firstly, due to the retrospective study design, certain informative data, such as secondary dengue infection, were unavailable. Secondly, our data only investigate SD patients and cannot be generalized to the overall dengue population, including patients in groups A or B as defined by the WHO [8]. Thirdly, there is a lack of information regarding the standard supportive therapy, which may introduce bias into the outcomes of SD patients.

In summary, we believe that instead of prioritizing patients with SD for intensive care treatment, it is crucial to identify independent risk factors such as old age, altered consciousness, gastrointestinal bleeding, and leukocytosis during the initial hospital presentation. Timely and prompt treatment based on these identified risk factors can play a crucial role in mitigating the severity of SD and reducing mortality rates. The manifestation of organ impairment during hospitalization indicates the advanced stage of dengue disease. Therefore, early recognition of signs of organ hypoperfusion and implementation of intensified supportive management to prevent further organ hypoperfusion should be prioritized as important strategies to decrease mortality.

## Supporting information

**S1 Data. Key laboratory findings in non-survivors.**
(XLSX)

**S2 Data. Key laboratory findings in survivors.**
(XLSX)

## Acknowledgments

We thank the medical staff of the Kaohsiung Chang Gung Memorial Hospital and Kaohsiung Medical University Hospital for the management of patients.

## Author Contributions

**Conceptualization:** Cheng-Hsun Yang, Ing-Kit Lee, Yen-Hsu Chen.

**Data curation:** Cheng-Hsun Yang, Ing-Kit Lee, Yi-Chun Chen, Wen-Chi Huang, Jui-Chi Hsu, Chien-Hsiang Tai, Chung-Hao Huang, Chun-Yu Lin, Yen-Hsu Chen.

**Funding acquisition:** Ing-Kit Lee.

**Investigation:** Ing-Kit Lee, Yen-Hsu Chen.

**Methodology:** Cheng-Hsun Yang, Ing-Kit Lee, Yi-Chun Chen, Yen-Hsu Chen.

**Project administration:** Ing-Kit Lee.

**Supervision:** Ing-Kit Lee, Yen-Hsu Chen.

**Validation:** Cheng-Hsun Yang, Ing-Kit Lee, Yi-Chun Chen, Wen-Chi Huang, Jui-Chi Hsu, Chien-Hsiang Tai, Chung-Hao Huang, Chun-Yu Lin, Yen-Hsu Chen.

**Visualization:** Ing-Kit Lee, Yen-Hsu Chen.

**Writing – original draft:** Cheng-Hsun Yang.

**Writing – review & editing:** Ing-Kit Lee.

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
