## [Decision Letter · Decision Letter 0]

15 Mar 2024

Dear Dr Lee,

Thank you very much for submitting your manuscript "Prognostic factors in severe dengue patients: A multi-center retrospective cohort study" for consideration at PLOS Neglected Tropical Diseases. As with all papers reviewed by the journal, your manuscript was reviewed by members of the editorial board and by several independent reviewers. In light of the reviews (below this email), we would like to invite the resubmission of a significantly-revised version that takes into account the reviewers' comments. 

We cannot make any decision about publication until we have seen the revised manuscript and your response to the reviewers' comments. Your revised manuscript is also likely to be sent to reviewers for further evaluation.

Sincerely,

Abdallah M. Samy, PhD

Section Editor

Aaron Jex

Section Editor

Reviewer's Responses to Questions

**Key Review Criteria Required for Acceptance?**

**Methods**

-Are the objectives of the study clearly articulated with a clear testable hypothesis stated?

-Is the study design appropriate to address the stated objectives?

-Is the population clearly described and appropriate for the hypothesis being tested?

-Is the sample size sufficient to ensure adequate power to address the hypothesis being tested?

-Were correct statistical analysis used to support conclusions?

-Are there concerns about ethical or regulatory requirements being met?

Reviewer #1: This is an interesting study because it is a disease with a major impact on public health and without a vaccine widely used in endemic countries. However, some aspects need to be improved before this manuscript is accepted for publication, which I will list below.

Improvement in the definition of inclusion and exclusion criteria: it is not clear whether only critically ill patients were included or whether patients with warning signs were included "The warning signs, as defined by the 2009 WHO, include abdominal pain or tenderness, persistent vomiting, clinical fluid accumulation, mucosal bleeding, and lethargy/restlessness. However, we were unable to include hepatomegaly and an increase in hematocrit concurrent

with a rapid decrease in platelet count as warning signs in our analysis due to the lack of information. SD was defined based on the 2009 WHO criteria, which categorizes it as dengue"

Characterization of the patient sample: How were they selected? Are these hospitals reference hospitals, but do they also serve patients through spontaneous demand?

Reviewer #2: The objectives of the study are clearly articulated, and the study design is appropriate to address the stated objectives. However, I suggest the inclusion of some missing information to ensure the reliability of the data and reproducibility.

Patients and Setting:

-The authors described the criteria to confirm the dengue cases included in the study. I suggest adding a brief explanation of how each of the assays was conducted and/or providing a reference. 

-Additionally, the primer sequences used in the RT-PCR should be included, along with the cycling parameters of PCR, the controls of the reaction, the machine used for running, and the criteria to consider a sample as positive (e.g., ct value).

-Was the detection of IgM and IgG performed using a kit? If so, how was this detection performed? Please clarify these methodologies, and if applicable, add the kit or reagents used.

Definitions:

-The authors stated that "severe hepatitis was defined as serum alanine aminotransferase (ALT) levels greater than 1,000 U/L (reference value, <40 U/L)." It would be clearer if the authors added why this cutoff of ALT levels was considered and provided a reference for it.

**Results**

-Does the analysis presented match the analysis plan?

-Are the results clearly and completely presented?

-Are the figures (Tables, Images) of sufficient quality for clarity?

Reviewer #1: (No Response)

Reviewer #2: The analysis presented matches the analysis plan. However, the presentation of results could be improved. 

- I suggest presenting the results comparing survivors and non-survivors in a histogram graph. This approach would provide a more visual representation and assist readers in discerning the differences between the groups, along with the accompanying statistical analysis.

- Additionally, if the data regarding the characteristics of fatal patients is presented in table 2, it should be referenced in the text. Therefore, Table 2 should be addressed in the text accordingly.

**Conclusions**

-Are the conclusions supported by the data presented?

-Are the limitations of analysis clearly described?

-Do the authors discuss how these data can be helpful to advance our understanding of the topic under study?

-Is public health relevance addressed?

Reviewer #1: (No Response)

Reviewer #2: The discussion and conclusions effectively support the presented data, addressing the public health relevance and discussing how the data can be beneficial. Additionally, the study limitations are appropriately addressed. However, to ensure robustness, please add references for the following sentences:

“These immunological changes further contribute to the risk of developing SD and mortality in this aging population.”

 “The WHO has documented that mucosal bleeding and the presence of lethargy or restlessness are pivotal warning signs for the development of SD”

“Moreover, severe plasma leakage in SD may lead to reduced cerebral perfusion, compromising the proper functioning of the brain, especially in susceptible populations such as elderly patients with multiple comorbidities.”

“Gastrointestinal bleeding can lead to hypovolemia, circulatory collapse, and shock. Delayed and inadequate fluid replacement, as well as insufficient transfusion of blood products in cases of severe bleeding, contribute to the development of circulatory failure and increased mortality.”

“The presence of organ impairment during hospitalization is another crucial factor contributing to mortality in patients with SD”

“Prolonged hypotension and shock resulting from severe plasma leakage and/or bacterial co-infection in cases of SD could lead to organ impairment such as liver and kidney damage. Moreover, the development of pneumonia indicates nosocomial infection, which can exacerbate the dengue disease and contribute to fatal outcomes.”

**Editorial and Data Presentation Modifications?**

Reviewer #1: (No Response)

Reviewer #2: (No Response)

**Summary and General Comments**

Reviewer #1: (No Response)

Reviewer #2: The introduction lacks essential information to contextualize the methods used and the obtained results. It would be beneficial to enrich the introduction to provide more context. For example, the authors could address concepts in the introduction that were not discussed but are pertinent to the results, such as "Dengue serotype 2." I suggest adding a section to the introduction describing dengue serotypes.

Additionally, the authors should verify and correct the writing of the virus name throughout the text.
---

## [Decision Letter · Decision Letter 1]

17 Sep 2024

Dear Dr Lee,

Thank you very much for submitting your manuscript "Prognostic factors in severe dengue patients: A multi-center retrospective cohort study" for consideration at PLOS Neglected Tropical Diseases. As with all papers reviewed by the journal, your manuscript was reviewed by members of the editorial board and by several independent reviewers. The reviewers appreciated the attention to an important topic. Based on the reviews, we are likely to accept this manuscript for publication, providing that you modify the manuscript according to the review recommendations. 

Sincerely,

Abdallah M. Samy, PhD

Section Editor

Reviewer's Responses to Questions

**Key Review Criteria Required for Acceptance?**

**Methods**

-Are the objectives of the study clearly articulated with a clear testable hypothesis stated?

-Is the study design appropriate to address the stated objectives?

-Is the population clearly described and appropriate for the hypothesis being tested?

-Is the sample size sufficient to ensure adequate power to address the hypothesis being tested?

-Were correct statistical analysis used to support conclusions?

-Are there concerns about ethical or regulatory requirements being met?

Reviewer #2: Which normality test was used before applying the Chi-square test or Fisher's exact test, the Student t test, or the Mann-Whitney U test? Please include this information to ensure the correct application of the tests.

**Results**

-Does the analysis presented match the analysis plan?

-Are the results clearly and completely presented?

-Are the figures (Tables, Images) of sufficient quality for clarity?

Reviewer #2: I understand that the results present many variables, which may make it difficult to display these data in a single graph. However, I still recommend creating graphs for at least the comparisons of the statistically significant variables. You could create a graph for each type of variable, for example: Graph A for the statistically significant data from the variable "Symptoms and signs upon arrival," Graph B for "Laboratory results upon arrival," and so on. This way, the graphical presentation would not duplicate the extensive information in the tables but would highlight the "most important" ones, making it clearer for the reader and further improving the overall quality of the article.

**Conclusions**

-Are the conclusions supported by the data presented?

-Are the limitations of analysis clearly described?

-Do the authors discuss how these data can be helpful to advance our understanding of the topic under study?

-Is public health relevance addressed?

Reviewer #2: (No Response)

**Editorial and Data Presentation Modifications?**

Reviewer #2: (No Response)

**Summary and General Comments**

Reviewer #2: The authors have made the modifications and included the points I suggested in the first review. The article is now more robust and reliable. I believe that including the statistical test used to verify the normal distribution of the data and attempting to create the suggested graphs would significantly improve the article.

PLOS authors have the option to publish the peer review history of their article (what does this mean?). If published, this will include your full peer review and any attached files.

Reviewer #2: Yes: Naiara Clemente Tavares

Figure Files:

Data Requirements:

Reproducibility:

References

---

## [Decision Letter · Decision Letter 2]

20 Nov 2024

PNTD-D-23-01029R2Prognostic factors in severe dengue patients: A multi-center retrospective cohort studyPLOS Neglected Tropical Diseases Dear Dr. Lee, Thank you for submitting your manuscript to PLOS Neglected Tropical Diseases. After careful consideration, we feel that it has merit but does not fully meet PLOS Neglected Tropical Diseases's publication criteria as it currently stands. Therefore, we invite you to submit a revised version of the manuscript that addresses the points raised during the review process. Please submit your revised manuscript within 30 days Dec 20 2024 11:59PM. If you will need more time than this to complete your revisions, please reply to this message or contact the journal office at plosntds@plos.org. Please include the following items when submitting your revised manuscript: *
A rebuttal letter that responds to each point raised by the editor and reviewer(s). You should upload this letter as a separate file labeled 'Response to Reviewers'. This file does not need to include responses to any formatting updates and technical items listed in the 'Journal Requirements' section below. *
A marked-up copy of your manuscript that highlights changes made to the original version. You should upload this as a separate file labeled 'Revised Manuscript with Track Changes'. *
An unmarked version of your revised paper without tracked changes. You should upload this as a separate file labeled 'Manuscript'. If you would like to make changes to your financial disclosure, competing interests statement, or data availability statement, please make these updates within the submission form at the time of resubmission. Guidelines for resubmitting your figure files are available below the reviewer comments at the end of this letter. We look forward to receiving your revised manuscript. Kind regards,Abdallah M. Samy, PhDSection EditorPLOS Neglected Tropical Diseases Aaron JexAssociate EditorPLOS Neglected Tropical Diseases

Shaden Kamhawi

co-Editor-in-Chief

Paul Brindley

co-Editor-in-Chief

**Additional Editor Comments :**Your manuscript received one additional review. The review is an statistical concern that should be addressed before considering a revised version of your manuscript. **Journal Requirements:**

1) We have noticed that you have uploaded Supporting Information files, but you have not included a list of legends. Please add a full list of legends for your Supporting Information files after the references list.

**Reviewers' comments:** Reviewer's Responses to Questions

**Key Review Criteria Required for Acceptance?**

**Methods**

-Are the objectives of the study clearly articulated with a clear testable hypothesis stated?

-Is the study design appropriate to address the stated objectives?

-Is the population clearly described and appropriate for the hypothesis being tested?

-Is the sample size sufficient to ensure adequate power to address the hypothesis being tested?

-Were correct statistical analysis used to support conclusions?

-Are there concerns about ethical or regulatory requirements being met?

Reviewer #2: (No Response)

**Results**

-Does the analysis presented match the analysis plan?

-Are the results clearly and completely presented?

-Are the figures (Tables, Images) of sufficient quality for clarity?

Reviewer #2: (No Response)

**Conclusions**

-Are the conclusions supported by the data presented?

-Are the limitations of analysis clearly described?

-Do the authors discuss how these data can be helpful to advance our understanding of the topic under study?

-Is public health relevance addressed?

Reviewer #2: (No Response)

**Editorial and Data Presentation Modifications?**

Reviewer #2: (No Response)

**Summary and General Comments**

Reviewer #2: The authors stated that “Additionally, the Mann-Whitney U test is a non-parametric test, meaning it can be used for data that does not follow a normal distribution” and “As the Mann-Whitney U test is non-parametric, it also does not require a normality test.” This is an absolutely incorrect statement. To determine if data follows a normal distribution, a normality test must be performed. It is not up to the author to decide whether the data has a normal distribution or not. The normality test should be conducted to verify this, after which one can select whether to use a parametric or a non-parametric test. Also, a non-parametric test does not imply that data lacks a normal distribution; rather, if data does not follow a normal distribution, a non-parametric test is used.

The authors should conduct a normality test to verify if the data follows a normal distribution and include this information for verification.

PLOS authors have the option to publish the peer review history of their article (what does this mean?). If published, this will include your full peer review and any attached files.

Reviewer #2: **Yes: **Naiara Clemente Tavares

---

## [Decision Letter · Decision Letter 3]

16 Jan 2025

Dear Dr Lee,

We are pleased to inform you that your manuscript 'Prognostic factors in severe dengue patients: A multi-center retrospective cohort study' has been provisionally accepted for publication in PLOS Neglected Tropical Diseases.

Best regards,

Abdallah M. Samy, PhD

Section Editor

Aaron Jex

%CORR_ED_EDITOR_ROLE%

Shaden Kamhawi

co-Editor-in-Chief

Paul Brindley

co-Editor-in-Chief

---

## [Editor Report · Acceptance letter]

23 Jan 2025

Dear Dr Lee,

We are delighted to inform you that your manuscript, "Prognostic factors in severe dengue patients: A multi-center retrospective cohort study," has been formally accepted for publication in PLOS Neglected Tropical Diseases.

Best regards,

Shaden Kamhawi

co-Editor-in-Chief

Paul Brindley

co-Editor-in-Chief
